# The Adult Congenital Heart Disease Anatomic and Physiological Classification: Associations with Clinical Outcomes in Patients with Atrial Arrhythmias

**DOI:** 10.3390/diagnostics12020466

**Published:** 2022-02-11

**Authors:** Anastasios Kartas, Andreas S. Papazoglou, Diamantis Kosmidis, Dimitrios V. Moysidis, Amalia Baroutidou, Ioannis Doundoulakis, Stefanos Despotopoulos, Elena Vrana, Athanasios Koutsakis, Georgios P. Rampidis, Despoina Ntiloudi, Sotiria Liori, Tereza Mousiama, Dimosthenis Avramidis, Sotiria Apostolopoulou, Alexandra Frogoudaki, Afrodite Tzifa, Haralambos Karvounis, George Giannakoulas

**Affiliations:** 1First Department of Cardiology, AHEPA University Hospital, Aristotle University of Thessaloniki, St. Kiriakidi 1, 54636 Thessaloniki, Greece; tkartas@gmail.com (A.K.); anpapazoglou@yahoo.com (A.S.P.); diamkosmidis@gmail.com (D.K.); dimoysidis@gmail.com (D.V.M.); bamalia27@gmail.com (A.B.); doudougiannis@gmail.com (I.D.); vranaelena@yahoo.com (E.V.); athkoutsakis@gmail.com (A.K.); grampidi@outlook.com (G.P.R.); ntiloudid@gmail.com (D.N.); hkarvounis@hotmail.com (H.K.); 2Department of Pediatric and Adult Congenital Heart Disease, Onassis Cardiac Surgery Center, 17674 Athens, Greece; stefanosdespotopoulos@gmail.com (S.D.); sotiria.apostolopoulou@gmail.com (S.A.); 3Second Department of Cardiology, Attikon University Hospital, 12462 Athens, Greece; sotiria.liori@gmail.com (S.L.); afrogou@otenet.gr (A.F.); 4Department of Congenital Heart Disease, Mitera Childrens’ Hospital, 15123 Athens, Greece; trs.156@gmail.com (T.M.); dimosthenisavramidis@yahoo.gr (D.A.); atzifa@mitera.gr (A.T.)

**Keywords:** atrial arrhythmia, ACHD, congenital heart disease, quality of life, SF-36, AP-ACHD classification

## Abstract

The implications of the adult congenital heart disease anatomic and physiological classification (AP-ACHD) for risk assessment have not been adequately studied. A retrospective cohort study was conducted using data from an ongoing national, multicentre registry of patients with ACHD and atrial arrhythmias (AA) receiving apixaban (PROTECT-AR study, NCT03854149). At enrollment, patients were stratified according to Anatomic class (AnatC, range I to III) and physiological stage (PhyS, range B to D). A follow-up was conducted between May 2019 and September 2021. The primary outcome was a composite of death from any cause, any major thromboembolic event, major or clinically relevant non-major bleeding, or hospitalization. Cox proportional-hazards regression modeling was used to evaluate the risks for the outcome among AP-ACHD classes. Over a median 20-month follow-up period, 47 of 157 (29.9%) ACHD patients with AA experienced the composite outcome. Adjusted hazard ratios (aHR) with 95% confidence intervals (CI) for the outcome in PhyS C and PhyS D were 1.79 (95% CI 0.69 to 4.67) and 8.15 (95% CI 1.52 to 43.59), respectively, as compared with PhyS B. The corresponding aHRs in AnatC II and AnatC III were 1.12 (95% CI 0.37 to 3.41) and 1.06 (95% CI 0.24 to 4.63), respectively, as compared with AnatC I. In conclusion, the PhyS component of the AP-ACHD classification was an independent predictor of net adverse clinical events among ACHD patients with AA.

## 1. Introduction

The advances of modern medicine have extended the survival of patients with congenital heart disease (CHD) well into adulthood. As these patients live longer, the focus of care expectantly shifts towards indexes of morbidity [1]. Mortality and morbidity in adults with congenital heart disease (ACHD) are profoundly affected by atrial arrhythmias (AA), which have a lifetime incidence of up to 15% [2,3,4,5,6]. Hence, reliable risk assessment in ACHD with AA is necessary. This process can be challenging, owing to the multifactorial aspects that affect prognosis in these patients.

Traditionally, risk in CHD has been thought to parallel the anatomic complexity of the respective lesion. Anatomic complexity was originally described in 2001 by the Bethesda Task Force [7]. Since it relies solely on anatomy, that classification scheme does not reflect the hemodynamic, arrhythmic, or functional disorders which may complicate CHD. Furthermore, the anatomic classification has been only modestly associated with mortality and morbidity [7,8,9]. In 2018, the American College of Cardiology and the American Heart Association proposed a refined scheme to address these shortcomings, namely the anatomic and physiological classification (AP-ACHD). The AP-ACHD classification combines information on the underlying CHD anatomic class (AnatC) with information regarding the patient’s physiological stage (PhyS) [10,11]. More detailed phenotyping of ACHD disease severity could enable structured risk stratification and tailored clinical decision-making. Despite the potential of this new classification, little data exist on its association with adverse outcomes, especially in patients with comorbid AA [12,13,14].

This study analyzed data from an ongoing multicentre registry of patients with ACHD and AA on oral anticoagulation with apixaban. The objective was to investigate the association of AP-ACHD classification with morbidity and mortality.

## 2. Materials and Methods

### 2.1. Study Design

This retrospective, observational cohort study included patients enrolled in the PROTECT-AR registry (PReventiOn of ThromboEmbolism in Adults with Congenital HearΤ disease and Atrial aRrhythmias, ClinicalTrials.gov Identifier: NCT03854149) between July 2019 and August 2021. PROTECT-AR is an ongoing, prospective registry that evaluates the safety and efficacy of apixaban treatment in the setting of ACHD with AA. The study is being conducted at four centers in the two largest cities in Greece, which serve the majority of the country’s ACHD population. A detailed protocol of the study was previously published [15]. Approval was obtained by the appropriate independent ethics committees or institutional review boards at each institution. Eligible inpatients or outpatients were consecutively enrolled after signing informed consent.

### 2.2. Study Population

For the purpose of this analysis, the study population included all patients enrolled in the PROTECT-AR study for whom baseline AP-ACHD classification and follow-up data were available. Patients were 18 years of age or older with documented ACHD and AA (known or new-onset atrial fibrillation, atrial flutter, or intra-atrial re-entrant tachycardia) and were treated with apixaban (i.e., new or ongoing users) for primary or secondary stroke prevention. Patients were prescribed apixaban at the discretion of their treating physicians.

### 2.3. Outcomes

The primary outcome in this study was the composite of mortality and morbidity occurring after the index date of enrollment in the study cohort. Mortality was defined as death from any cause. Morbidity included any major thrombotic event, any major or clinically relevant non-major bleeding event, or hospitalization for any cause. Major thrombotic events included ischemic stroke, systemic or pulmonary embolism, intracardiac thrombus, or myocardial infarction. Bleeding events were stratified according to the International Society on Thrombosis and Hemostasis Guidelines [16]. The criteria for the coding of events are summarized in Appendix A [15].

### 2.4. Data Sources and Measures

Patient data regarding ACHD type, AnatC, and PhyS components of the AP-ACHD classification, type of AA, demographics, clinical characteristics, medical history, laboratory, electrocardiographic, echocardiographic data, and health status metrics were assessed at enrollment as part of the PROTECT-AR study. The AP-ACHD classification was assigned by ACHD expert physicians who used the consensus definitions (Appendix A) [10]. In the AP-ACHD scheme, AnatC is stratified as I (simple), II (moderate), or III (great complexity); PhyS ranges from A (no hemodynamic, functional, anatomic, or arrhythmic sequelae) to D (severe sequelae). The most severe clinical features dictate each of the respective AnatC and PhyS. Since AA (i.e., presence of arrhythmic sequelae) was featured among all study participants as per the study’s inclusion criteria, PhyS was assigned a classification B at minimum. The combination of AnatC and PhyS composed the AP-ACHD classification.

Baseline health status was assessed by the modified European Heart Rhythm Association (mEHRA) classification and Short Form-36 (SF-36) scores. These were completed via paper-and-pen versions in-person. The mEHRA score gauges arrhythmia-related symptom burden on the patient’s daily activity from none (class 1) to disabling (class 4) [17]. The SF-36 is a 36-item self-report questionnaire used to evaluate health-related quality of life (QoL) [18]. It evaluates physical functioning, limitations due to physical health problems, bodily pain, energy/fatigue, social functioning, limitations due to emotional problems, and psychological distress and well-being with scores ranging from 0 (worst) to 100 (best). These 8 domains can be combined into 2 higher-ordered clusters, the physical component summary and the mental component summary [19].

Follow-up data were collected 1 month after patients’ enrolment and then every 6 months thereafter by in-person or telephonic interviews. The last follow-up for this study occurred in September 2021.

### 2.5. Statistical Analysis

Descriptive statistics of the baseline patient characteristics were calculated overall and after stratifying by AnatC and PhyS. Data were summarized as means with standard deviation, medians with interquartile ranges, or frequencies and percentages. Trends across AP-ACHD classification classes were calculated by modeling the AnatC and PhyS of AP-ACHD as ordinal variables. Linear regression was used for continuous variables, and logistic or multinomial logistic regression was used for categorical variables.

Incidence rates for all outcomes were determined as events per 100 patient-years with associated 95% confidence intervals (CIs). Kaplan–Meier curves were plotted to describe the unadjusted association of AnatC and PhyS with the occurrence of the composite outcome. Multivariable Cox regression analysis was used to estimate the corresponding adjusted hazard ratios (aHRs). The groups with the least severe AnatC and PhyS were used as references. Adjusting required eight to ten composite events per each covariate entered into the model to avoid biased effect estimates [20]. Model covariates were selected a priori on the basis of incorporating adequate information on thrombotic and bleeding risk. These covariates were AnatC or PhyS, age, CHA2DS2-VASc, HAS-BLED scores, and left atrial (LA) diameter, as assessed at baseline. For patients who were lost to follow-up, all information collected to the point of final contact was analyzed. For patients who did not experience an outcome event, their time-to-event measure was censored at the last contact date. The SF-36 scores for each domain were standardized for each component summary using a z-score transformation [19] and aggregated using factor score coefficients.

Every statistical test was two-tailed, with statistical significance determined at the level of 0.05. Statistical analysis was performed using IBM SPSS Statistics version 24.0 (IBM Corp, Armonk, NY, USA), STATA 13 software (StataCorp. 2013. Stata Statistical Software: Release 13. College Station, TX, USA: StataCorp LP), and Python IDLE Shell 3.9.5.

## 3. Results

### 3.1. Population Characteristics

From May 2019 through August 2021, a total of 171 ACHD patients with AA under routine apixaban therapy were enrolled. The final cohort for this analysis included 157 patients, after the exclusion of 13 patients with missing data on AP-ACHD classification and 1 patient with missing follow-up data. The mean age was 51.5 years, and 53.8% were women. Atrial fibrillation was the most common AA (74.5%), and paroxysmal AA was the most common temporal pattern (46.5%). The median CHA2DS2-VASc score was 2 (IQR = 1.25), and the median HAS-BLED score was 1 (IQR = 2). The most common ACHD type was repaired tetralogy of Fallot (17.1%) (see the prevalence of ACHD types in Appendix A).

Most patients had moderate anatomic complexity (AnatC II, 35.7%) and mildly impaired physiology (PhyS B, 59.2%). Overall, AP-ACHD class IIB composed the majority of the cohort (21.7%). Table 1 and Table 2 show the baseline characteristics of patients according to AnatC and PhyS, respectively. Patients with more complex AnatC were significantly younger with lower systemic ventricular ejection fraction, higher prevalence of paroxysmal AA, and fewer cardiovascular risk factors (*p* values for trend < 0.05). Patients with worse PhyS were significantly older and had a higher prevalence of heart failure and chronic kidney disease. They also had more severe NYHA class (III/IV), increased LA size, and lower physical component summary score (*p* values for trend < 0.05). More severe AnatC and PhyS had a higher prevalence of intra-atrial reentry tachycardia and lower and higher mEHRA scores, respectively (*p* for trend < 0.05). AP-ACHD classes IC and ID had the highest mEHRA scores (2 and 2.5, respectively). In most AP-ACHD classes, the mean physical and mental component summaries of the SF-36 score ranged below 50%. A visual distribution of the AP-ACHD classes and the associated mEHRA and SF-36 scores at baseline is presented in Appendix A.

### 3.2. AP-ACHD Classification and Adverse Events

Over a median 20-month follow-up (IQR = 15 months), 47 (29.9%) patients experienced at least one component of the composite outcome. In particular, 6 patients died from any cause, 3 experienced a major bleeding event, 19 experienced a clinically relevant non-major bleeding event, and 19 were hospitalized for any cause. No patient sustained a thromboembolic event. Kaplan–Meier-derived probability of the composite outcome occurring among AP-ACHD classes is depicted in Figure 1. That probability ranged from 20 (class IIIB) to 50% (classes ID and IIC) during the follow-up period. In Figure 2, the cumulative incidence curves of the composite outcome within AnatC and PhyS subgroups overlap. This indicated non-significant differences in univariate analysis (incidence rates and unadjusted HRs within AnatC and PhyS are seen in Appendix A). After adjustment for AnatC, age, CHA2DS2-VASc, HAS-BLED scores, and LA diameter, the PhyS D was associated with increased risk of the composite outcome (all-cause mortality, thromboembolism, major or clinically relevant non-major bleeding, and hospitalization), as compared with PhyS B (aHR: 8.15, CI: 1.52 to 43.59; *p* = 0.014). The rest of the aHRs concerning AnatC and PhyS subgroups remained non-significant. The incidence rates and HRs of the individual components of the composite outcome did not differ significantly among AnatC and PhyS (Appendix A).

## 4. Discussion

In this retrospective analysis of a prospectively followed-up cohort of ACHD patients with AA on apixaban, the PhyS component of the AP-ACHD classification was independently associated with a composite of fatal and major non-fatal adverse outcomes. In particular, PhyS D was associated with an almost eight-fold increased risk of the composite outcome occurrence over an approximate 2-year follow-up period, as compared with PhyS B. The significance of this association emerged after adjusting for clinically relevant confounders, including AnatC, age, CHA2DS2-VASc and HAS-BLED scores, and LA size. On the other hand, the AnatC component was not predictive of adverse clinical outcomes. To the best of our knowledge, this is the first study to investigate the prognostic implications of the recently proposed AP-ACHD classification in patients with ACHD and AA.

Our cohort mainly consisted of middle-aged ACHD patients with moderate anatomic complexity and mildly impaired physiology. Comorbidities were common, especially in more impaired physiologies. Health status metrics (quality of life, arrhythmia-related symptoms) ranged poorly. Furthermore, one out of five patients sustained a fatal or major non-fatal event during a median 20-month follow-up. This is notable, given the majority were enrolled as stable outpatients. These data underscore the additional burden conferred by AA in ACHD and call for improved risk stratification in that setting.

Our results concur with the published literature of similarly designed studies in the broader ACHD setting, whereby PhyS was superior to AnatC in risk assessment [21,22,23]. Ombelet et al. were the first to describe the added prognostic value of PhyS alongside AnatC in the AP-ACHD classification. Nevertheless, their study only included mortality as an outcome measure [23]. A study of 1000 ACHD outpatients found a modest improvement in the discriminative ability of anatomic classification by the addition of the PhyS. Stronger predictive models were created by including biomarkers, such as NT-proBNP [21]. Anatomic complexity was not included in a parsimonious prediction model of morbidity and mortality, which was developed by Geenen et al. [22]. Of PhyS parameters assessed (NYHA class, oxygen saturation < 90%, AA, ventricular or valvular dysfunction), only NYHA class retained significance in the multivariable model.

The current study provides insight into a more specialized ACHD population with comorbid AA on oral anticoagulation treatment. As a side note, this study serves as a reminder of two of the major challenges in outcomes-based research in ACHD: heterogeneity and lack of sizeable study populations. Adding functional on top of anatomic classification creates a multilevel classification scheme that might further complicate the process of estimating risk, simply due to the scarcity of available data to base risk estimates on. Our results suggest that PhyS rather than AnatC might be a more important driver of adverse sequelae in ACHD with comorbid AA. Conventional wisdom could explain this observation: individual parameters assessed when scoring PhyS, such as heart failure, pulmonary hypertension, or valvular heart disease, possess an independent prognostic value in either ACHD or AA [24,25,26,27]. On the other hand, anatomic complexity does not convey any information on the patient’s hemodynamic, functional status, or comorbidities; a patient with atrial septal defect and severe pulmonary hypertension could still be classified as simple anatomic complexity. Although our findings should be interpreted with caution, they may pave the way to more comprehensive risk assessment in ACHD with AA. Indeed, more data comparing AnatC, PhyS, and novel prediction models are needed. If PhyS does prove to be a reliable indicator of prognosis, the next step would be to assess whether efforts towards improving PhyS lead to better clinical outcomes.

## 5. Limitations

This analysis was not planned at the outset of the PROTECT-AR study. Hence, the results should be seen as secondary findings of an observational study. Classification of AP-ACHD is often subject to substantial interobserver variability [28]. Extensive adjustment for confounders was precluded due to a combination of sample size and event restrictions, whereas unmeasured confounding cannot be excluded. Median follow-up was limited, albeit comparable to studies of similar scope. Furthermore, results may not be generalizable to ACHD patients with AA being treated with other OAC apart from apixaban or even to populations across different countries as inter-country variation affects patient-reported outcomes in ACHD [29]. Finally, the restricted sample size, especially in the PhyS D subgroup, did not permit generalizing results for each distinctive AP-ACHD class.

## 6. Conclusions

In this observational cohort of patients with ACHD and AA, physiological impairment, as expressed by the PhyS component of the AP-ACHD classification, was an independent predictor of major adverse events. Further research on predictive modeling of adverse outcomes is needed to inform better management of the care of patients with ACHD and AA.

## Figures and Tables

**Figure 1 diagnostics-12-00466-f001:**
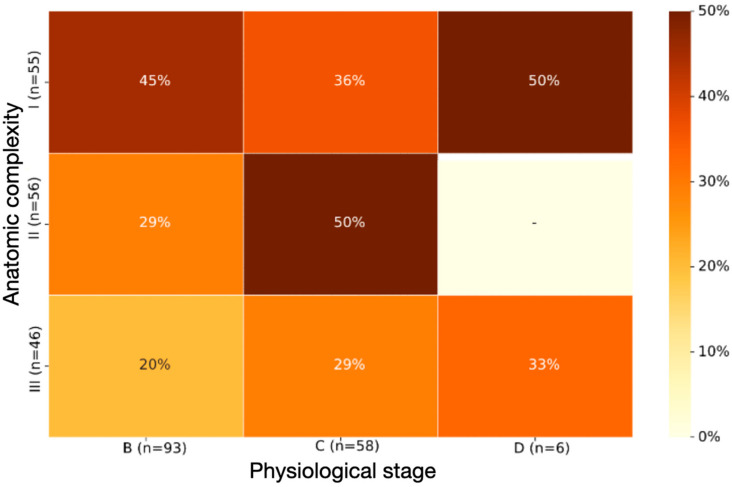
Probability of the composite outcome for various AP-ACHD classes based on Kaplan–Meier estimates. Kaplan–Meier estimates derived from maximum likelihood estimation of hazard function for each separate AP-ACHD class. For instance, the probability of the composite outcome occurring to patients in the IIIB AP-ACHD class is 20% at 20 months of follow-up (median follow-up duration). The cohort did not include any patient classified as IID. Hence, the probability of the outcome cannot be estimated for this subgroup. AP-ACHD, anatomic and physiological classification of adult congenital heart disease.

**Figure 2 diagnostics-12-00466-f002:**
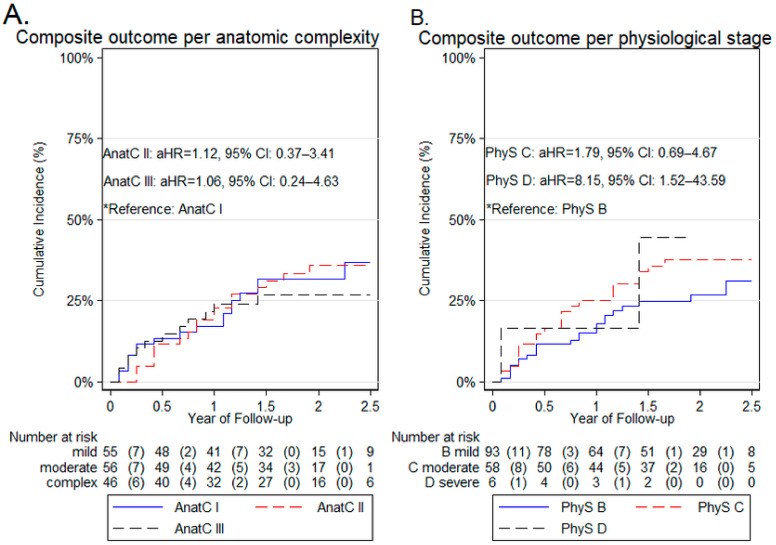
Cumulative incidence curves of the composite outcome by (**A**) AnatC, and (**B**) PhyS. Cumulative incidence curves of the composite outcome by (**A**) AnatC and (**B**) PhyS. AnatC, anatomic class; PhyS, physiological stage.

**Table 1 diagnostics-12-00466-t001:** Descriptive and Clinical Characteristics by Anatomic Class.

	I (*n* = 55)	II (*n* = 56)	III (*n* = 46)	*p*-Value for Trend
**Clinical characteristics**				
**Age (years), mean ± SD**	63.9 ± 12.5	49.9 ± 13.6	38.8 ± 14.5	**<0.001**
**Female gender**	30 (54.5%)	31 (55.4%)	24 (52.2%)	0.766
**BMI (kg/m^2^), mean ± SD**	27.4 ± 4.8	28.3 ± 6.1	23.9 ± 3.2	**0.012**
**Systolic blood pressure (mmHg), mean ± SD**	122 ± 18	120 ± 17	113 ± 13	**0.019**
**Diastolic blood pressure (mmHg), mean ± SD**	76 ± 12	70 ± 8	70 ± 9	**0.006**
**AP-ACHD Physiological Stage**				
**B**	30 (54.5%)	34 (60.7%)	29 (63%)	0.975
**C**	22 (40%)	22 (39.3%)	14 (30.5%)	0.692
**D**	3(5.5%)	0 (0%)	3 (6.5%)	0.860
**CHA2DS2-VASc score, median (IQR)**	2 (6)	1 (4)	1 (3)	**0.002**
**HAS-BLED score, median (IQR)**	1 (5)	0 (3)	1 (3)	**0.008**
**Medical history**				
**Stroke/thromboembolism**	6 (10.9%)	2 (3.6%)	9 (19.6%)	0.130
**Major bleeding**	2 (3.6%)	2 (3.6%)	1 (2.2%)	0.658
**Clinically relevant non-major bleeding**	8 (14.5%)	7 (12.5%)	4 (8.7%)	0.338
**Minor bleeding episodes per month, median (IQR)**	0 (4)	0 (4)	0 (4)	0.768
**Heart failure**	24 (43.6%)	27 (48.2%)	23 (50%)	0.474
**NYHA functional class**				
**I**	15 (32.6%)	14 (30.4%)	15 (37.5%)	**<0.001**
**II**	17 (37%)	24 (52.2%)	18 (45%)	**<0.001**
**III/IV**	14 (30.4%)	8 (17.4%)	7 (17.5%)	0.146
**NT-pro-BNP (pg/mL), median (IQR)**	2043 (44250)	772 (16763)	364 (4379)	**0.019**
**Smoking**	8 (14.5%)	2 (3.6%)	1 (2.2%)	**0.013**
**Dyslipidemia**	16 (29.1%)	10 (17.9%)	0 (0%)	**<0.001**
**Arterial Hypertension**	23 (41.8%)	14 (25%)	3 (6.5%)	**<0.001**
**Diabetes mellitus**	9 (16.4%)	11 (19.6%)	2 (4.3%)	0.112
**Chronic kidney disease**	3 (5.5%)	3 (5.4%)	3 (6.5%)	0.888
**Catheter ablation**	5 (9.1%)	3 (5.4%)	3 (6.5%)	0.640
**Atrial Arrhythmia**				
**Atrial fibrillation**	49 (89.1%)	43 (76.8%)	26 (56.5%)	**<0.001**
**Atrial flutter**	4 (7.3%)	10 (17.9%)	10 (21.7%)	**0.032**
**Intraatrial reentry tachcardia**	2 (3.6%)	3 (5.4%)	10 (21.7%)	**<0.001**
**First-diagnosed**	4 (8.6%)	4 (8.3%)	2 (6.5%)	0.457
**Paroxysmal**	24 (51%)	25 (52.1%)	24 (77.4%)	**0.013**
**Persistent or permanent**	19 (40.4%)	19 (39.6%)	5 (16.1%)	0.098
**Years since diagnosis, median (IQR)**	5 (30)	4 (32)	5 (15)	0.834
**QRS duration (msec), mean ± SD**	105 ± 20	126 ± 22	115 ± 27	0.364
**Echocardiography**				
**Systemic ventricular fraction (%), mean ± SD**	55 ± 7	54 ± 8	47 ± 8	**<0.001**
**LA diameter (cm), mean ± SD**	5 ± 0.9	4.7 ± 1.1	4.3 ± 0.6	**0.013**
**Health status metrics**				
**mEHRA score, median (IQR)**	2 (4)	1 (3)	1 (3)	**0.020**
**Physical component summary of SF-36, median (IQR)**	47.4 (74)	47.3 (32)	45.2 (86)	0.763
**Mental component summary of SF-36, median (IQR)**	47.6 (96)	48.3 (48)	49.6 (66)	0.897
**Medication**				
**Class I antiarrhythmic**	4 (7.3%)	5 (8.9%)	1 (2.2%)	0.384
**Class III antiarrhythmic**	10 (18.2%)	18 (32.1%)	13 (28.3%)	0.230
**beta-blocker**	30 (66.7%)	33 (64.7%)	19 (47.5%)	0.390

*p*-values < 0.05 (bold) were considered statistically significant. AP-ACHD, anatomic and physiological classification of adult congenital heart disease; BMI, body mass index; CHA2DS2-Vasc (congestive heart failure, hypertension, age ≥ 75 years, diabetes mellitus, stroke/transient ischemic attack, vascular disease, age 65 to 74 years, sex category); HAS-BLED (Hypertension, Abnormal renal/liver function, Stroke, Bleeding history or predisposition, Labile INR, Elderly, Drugs/alcohol); IQR, interquartile range; NT-proBNP, N-terminal pro-BNP; mEHRA, modified European Heart Rhythm Association; SD, standard deviation; SF-36, Short Form-36.

**Table 2 diagnostics-12-00466-t002:** Descriptive and Clinical Characteristics by Physiological Stage.

	B (*n* = 93)	C (*n* = 58)	D (*n* = 6)	*p*-Value for Trend
**Clinical characteristics**				
**Age (years), mean ± SD**	48.8 ± 17	55.8 ± 15.2	52.3 ± 22.8	**0.017**
**Female gender**	51 (54.8%)	32 (55.2%)	2 (33.3%)	0.660
**BMI (kg/m^2^), mean ± SD**	26.6 ± 4.6	27.3 ± 6.1	23.6 ± 4.4	0.803
**Systolic blood pressure (mmHg), mean ± SD**	118 ± 14	120 ± 19	116 ± 12	0.703
**Diastolic blood pressure (mmHg), mean ± SD**	72 ± 10	72 ± 11	73 ± 8	0.625
**AP-ACHD Anatomic class**				
**I**	30 (32.3%)	22 (37.9%)	3 (50%)	0.110
**II**	34 (36.6%)	22 (37.9%)	0 (0%)	**<0.001**
**III**	29 (31.1%)	14 (24.1%)	3 (50%)	0.672
**CHA2DS2-VASc score, median (IQR)**	1 (6)	2 (5)	3 (3)	**0.027**
**HAS-BLED score, median (IQR)**	1 (3)	1 (5)	2.5 (4)	**0.028**
**Medical history**				
**Stroke/thromboembolism**	7 (7.5%)	4 (6.9%)	2 (33.3%)	0.748
**Major bleeding**	3 (3.2%)	2 (3.4%)	0 (33.3%)	0.878
**Clinically relevant non-major bleeding**	10 (10.8%)	8 (13.8%)	1 (16.6%)	0.601
**Minor bleeding episodes per month, median (IQR)**	0 (4)	0 (4)	1 (2)	0.242
**Heart failure**	33 (35.5%)	37 (63.8%)	4 (66.6%)	**0.004**
**NYHA functional class**				
**I**	35 (47.9%)	8 (15.1%)	1 (16.7%)	0.292
**II**	29 (39.7%)	27 (50.9%)	3 (50%)	0.914
**III/IV**	9 (12.3%)	18 (33.9%)	2 (33.3%)	**<0.001**
**NT-pro-BNP (pg/mL), median (IQR)**	507 (1927)	1006 (16611)	213 (44295)	**0.016**
**Smoking**	5 (5.4%)	4 (6.9%)	2 (33.3%)	0.226
**Dyslipidemia**	17 (18.3%)	8 (13.8%)	1 (16.6%)	0.359
**Arterial Hypertension**	25 (26.9%)	12 (20.7%)	2 (33.3%)	0.379
**Diabetes mellitus**	11 (11.8%)	9 (15.5%)	1 (16.6%)	0.734
**Chronic kidney disease**	1 (1.1%)	7 (12.1%)	1 (16.6%)	**0.011**
**Catheter ablation**	7 (7.5%)	3 (5.2%)	1 (16.6%)	0.145
**Atrial Arrhythmia**				
**Atrial fibrillation**	66 (71%)	48 (82.8%)	4 (66.7%)	0.068
**Atrial flutter**	17 (18.3%)	7 (12.1%)	0 (0%)	**0.003**
**Intraatrial reentry tachcardia**	10 (10.8%)	3 (5.2%)	2 (33.3%)	**0.001**
**First-diagnosed**	6 (8.9%)	3 (5.7%)	1 (16.6%)	0.128
**Paroxysmal**	45 (67.2%)	24 (46.2%)	3 (50%)	**0.002**
**Persistent or permanent**	16 (23.9%)	25 (48.1%)	2 (33.3%)	**0.022**
**Years since diagnosis, median (IQR)**	4 (32)	5 (28)	7 (23)	0.187
**QRS duration (msec), mean ± SD**	118 ± 25	117 ± 32	116 ± 22	0.919
**Echocardiography**				
**Systemic ventricular fraction (%), mean ± SD**	53 ± 7	49 ± 10	54 ± 4	0.052
**LA diameter (cm), mean ± SD**	4.4 ± 0.6	5.2 ± 1.2	5.3	**<0.001**
**Health status metrics**				
**mEHRA score, median (IQR)**	1 (4)	2 (4)	2 (3)	**0.001**
**Physical component summary of SF-36, median (IQR)**	49.6 (86.3)	41.6 (73.8)	44.9 (9.3)	**<0.001**
**Mental component summary of SF-36, median (IQR)**	51.7 (66.4)	43.1 (96.5)	42.4 (26.8)	**0.031**
**Medication**				
**Class I antiarrhythmic**	8 (8.6%)	2 (3.4%)	0 (0%)	0.179
**Class III antiarrhythmic**	25 (26.9%)	14 (24.1%)	2 (33.3%)	0.851
**beta-blocker**	45 (48.4%)	34 (58.6%)	3 (50%)	0.070

*p*-values < 0.05 (bold) were considered statistically significant. AP-ACHD, anatomic and physiological classification of adult congenital heart disease; BMI, body mass index; CHA2DS2-Vasc (congestive heart failure, hypertension, age ≥ 75 years, diabetes mellitus, stroke/transient ischemic attack, vascular disease, age 65 to 74 years, sex category); HAS-BLED (Hypertension, Abnormal renal/liver function, Stroke, Bleeding history or predisposition, Labile INR, Elderly, Drugs/alcohol); IQR, interquartile range; NT-proBNP, N-terminal pro-BNP; mEHRA, modified European Heart Rhythm Association; SD, standard deviation; SF-36, Short Form-36.

## Data Availability

The data presented in this study are available on request from the corresponding author.

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
