# Peer review of "The Adult Congenital Heart Disease Anatomic and Physiological Classification: Associations with Clinical Outcomes in Patients with Atrial Arrhythmias"

_diagnostics, 2022, doi:10.3390/diagnostics12020466_

Round 1

Reviewer 1 Report

Reviewer comment:

In this retrospective analysis, Kartas et al. sought to investigate the association of anatomic (AnatC) and physiological (PhyS) classification in adult patients with congenital heart disease (AP-ACHD) and atrial arrhythmia (AA) with morbidity and mortality and found that the PhyS component of the AP-ACHD classification was an independent predictor of net adverse clinical events among ACHD patients with AA receiving apixaban. Although this study may contain important clinical implications, there exists several limitations.

Specific comments:

#1.         The present study demonstrated that physiological dysfunction at baseline was associated with poor clinical outcomes during follow-up of 20 months in patients with ACHD and AA, which was relatively short duration for long-term observation due to the secondary findings of an original observational study. The result was concordant with the previous literature as cited by the authors (reference 23-25), thus not surprising. Furthermore, among the enrolled 157 patients, patients who were classified in phyS stage D was only 6 patients, in 2 of whom hospitalization and/or bleeding events occurred. Therefore, it is premature to draw the conclusions.

#2.         In table S4, HRs were adjusted for CHA2DS2-VASc and HAS-BLED scores and left atrial diameter. Considering the difference in anatomic classes among the patients with three physiological stages, HRs should also be adjusted by the anatomic classes.

#3.         In Figure 1, estimated probability of the composite outcome in AP-ACHD class IIID was lower than class IB, which was discordant with the overall conclusions. The statistical methods for the analysis and interpretation should be further described.

#4.         Despite the significant higher aHRs in patients with phyS stage D (Table S4), cumulative incidence curves of the composite outcome within PhyS were overlapping (Figure 2). As stated above, The HRs may be influenced by additional components (anatomic classes) for adjustment.

#5.         In this study, antiarrhythmic drugs were used for the treatment of AA in 27.1 to 35.5% of the patients in three phyS stages. Recent studies demonstrated that catheter ablation provided improvements in survival, freedom from arrhythmia recurrence, and quality of life relative to drug therapy in patients with heart failure. (1,2) The authors should clarify how many patients underwent catheter ablation for atrial fibrillation and/or atrial tachycardia/flutter.

  1. Packer DL, Piccini JP, Monahan KH et al. Ablation Versus Drug Therapy for Atrial Fibrillation in Heart Failure: Results From the CABANA Trial. Circulation 2021;143:1377-1390.
  2. Marrouche NF, Brachmann J, Andresen D et al. Catheter Ablation for Atrial Fibrillation with Heart Failure. The New England journal of medicine 2018;378:417-427.

Reviewer 2 Report

This is a retrospective clinical study from PROTECT-AR registry (PReventiOn of ThromboEmbolism in Adults with Congenital HearΤ disease and Atrial aRrhythmias), which is an ongoing, prospective registry that evaluates the safety and efficacy of apixaban treatment in the setting of ACHD with AA. The present study aimed to investigate the association of AP-ACHD classification with morbidity and mortality. The authors concluded that physiological impairment, as expressed by the PhyS component of the AP-ACHD classification, was an independent predictor of major adverse events in patients with ACHD and AA. This is an important issue, and this reviewer considers that the authors well performed the present study. This reviewer has some comments as described below.

Minor comments:

  1. Tables 1-2. “b-blocker” seems to be “beta-blocker” or “β-blocker”.
  2. Figure 2. When was the start of the follow-up? Start date of apixaban? If the start date is the enrolled day of PROTECT-AR registry, when apixaban was administered? The authors should clearly define the starting date of the present study and apixaban administration in the Methods section.

Round 2

Reviewer 1 Report

Thank you for the revision. The authors underwent appropriate revision according to the reviewers' suggestion and comments. The reviewer has no further comments.